# A Prospective, Real-World, Multinational Study of Naloxegol for Patients with Cancer Pain Diagnosed with Opioid-Induced Constipation—The NACASY Study

**DOI:** 10.3390/cancers14051128

**Published:** 2022-02-23

**Authors:** Andrew Davies, Saverio Cinieri, Denis Dupoiron, Sofia España Fernandez, Johan Leclerc, Vincenzo Montesarchio, Kyriaki Mystakidou, Judith Serna, Jan Tack

**Affiliations:** 1Trinity College Dublin, University College Dublin, Our Lady’s Hospice Dublin’, D6W RY72 Dublin, Ireland; 2Medical Oncology, ASL Brindisi–P.O. “A. Perrino”, 72100 Brindisi, Italy; saverio.cinieri@gmail.com; 3Département d’Anesthésie–Douleur, Institut de Cancerologie de l’Ouest–Site Paul Papin, 49055 Angers, France; denis.dupoiron@ico.unicancer.fr; 4Medical Oncology Department, Institut Català d’Oncologia-Badalona, 08916 Barcelona, Spain; sofia.ef@iconcologia.net; 5Doleur, CHU Amiens Picardie, 80054 Amiens, France; johan.leclerc@wanadoo.fr; 6UOC Ongologia, A.O.R.N. dei Colli–Monaldi–Coutgno–C.T.O. Hospitals, 80131 Napoli, Italy; vincenzo.montesarchio@ospedalideicolli.it; 7Palliative Care Unit “Jenny Karezi”, School of Medicine, University of Athens, 11526 Athens, Greece; mistakidou@yahoo.com; 8Hospital Universitari Vall D’hebron, 08035 Barcelona, Spain; jserna@vhebron.net; 9University Hospital Gasthuisberg, 3000 Leuven, Belgium; jan.tack@kuleuven.be

**Keywords:** naloxegol, PAMORA, MOVANTIK, opioid-induced constipation, cancer pain, real-world

## Abstract

**Simple Summary:**

Prescription opioid pain medications help ease pain, but they also cause some unwanted side effects such as constipation. In this study we evaluated the safety and efficacy of naloxegol used to treat constipation that is caused by opioids. We found that naloxegol improved constipation and quality of life in patients with cancer-related pain and opioid-induced constipation.

**Abstract:**

The Naloxegol Cancer Study (NACASY) was a multinational European study aimed to evaluate the 4-week safety and efficacy of naloxegol in a real-world setting in patients with cancer pain diagnosed with opioid-induced constipation. The primary safety endpoint was the incidence of adverse events leading to study discontinuation. We recruited 170 patients who received at least one dose of naloxegol (i.e., safety population). Out of 170 patients, 20 (11.8%, 95%CI 6.9–16.6) discontinued the study due to adverse events, and, of them, 12 (7.1%, 95%CI 3.2–10.9%) were study discontinuations due to naloxegol-related adverse events. From 76 patients subjects who had completed both 4 weeks of treatment and 28 days of the diary, 55 patients (72.4%, 95% CI 62.3–82.4%) were regarded as responders (i.e., showed ≥3 bowel-movements per week and an increase of ≥1 bowel-movement over baseline) to naloxegol treatment. The Patient Assessment of Constipation—Quality of Life Questionnaire total score and all its subscales improved from baseline to 4 weeks of follow up. Our findings support and provide new evidence about the beneficial effect of naloxegol in terms of improvement of constipation and quality-of-life in patients with cancer-related pain and opioid-induced constipation and show a safety profile consistent with previous pivotal and real-world studies.

## 1. Introduction

Cancer-related pain is one of the most frequent and bothersome symptoms affecting patients with cancer [1]. The frequency of cancer pain is related to the stage of disease. Pain prevalence increases from 33% in patients after curative treatment up to 64% in patients with metastatic, advanced, or terminal disease [2].

Opioids remain the cornerstone of analgesic treatment for severe cancer pain [3,4]. However, many opioid-associated adverse events may compromise their utilization [5]. Among these, opioid-induced constipation (OIC) is the most common and debilitating reported side-effect, with an overall prevalence ranging from 51% to 87% [6]. OIC negatively impacts patients’ health-related quality of life (HRQoL), and impairs patients’ ability to perform daily activities [7]. Hence, many patients decide to discontinue or limit their opioid therapy, resulting in insufficient pain control [8].

From a pathophysiological standpoint, OIC is predominantly the result of the agonist-binding and subsequent action of opioids on the peripheral network of μ-opioid receptors in the gastrointestinal tract [7,9]; this leads to a constellation of effects, such as impaired esophageal motility, slowing of gastric emptying, and extended intestinal transit times, reduced intestinal secretions, and increased tone in intestinal sphincters [6]. Therefore, OIC treatment strategies should specifically address opioid binding to the GI peripheral μ-opioid receptors [6].

Dietary modifications and lifestyle changes, followed by over-the-counter (OTC) stool softeners and laxatives, are the first-step management recommendations for OIC [7,10]. Nevertheless, the efficacy of OTC laxatives is limited, because they do not address the underlying pathophysiological cause of OIC [9,11,12]. Furthermore, in addition to lack of benefit [7,8,10], several laxative associated side-effects, such as bloating, nausea, fecal incontinence, dehydration, and electrolyte imbalance, are frequently reported [13,14]. 

Peripherally acting μ-opioid receptor antagonists (PAMORAs) constitute a novel class of drugs indicated for OIC that specifically bind and block the μ-opioid receptors of the gastrointestinal tract, without crossing the blood–brain barrier and, hence, without interacting with the central nervous system (CNS) opioid receptors. Therefore, they have the ability to alleviate OIC without compromising the opioid’s analgesic effects [11,12]. In patients who do not respond to standard laxatives, PAMORAs are a valid therapeutic option [15,16].

Naloxegol, a PEGylated derivate of naloxone, was the first orally administered PAMORA approved in 2014 for the treatment of OIC [17,18]. It is indicated for the treatment of OIC in noncancer and cancer adult patients with an inadequate response to laxatives; the recommended dosage is 25 mg once daily and, if this is not tolerated, the dose could be reduced to 12.5 mg once daily [18].

The clinical safety and efficacy of naloxegol was demonstrated in two identical phase III, 12-week, randomized, double-blind, placebo-controlled clinical trials, conducted in adult outpatients with noncancer pain and OIC (KODIAC-04 and KODIAC-05) [19]. Furthermore, the good safety and tolerability profile reported in these two studies was later corroborated in a 52-week trial in patients with noncancer pain and OIC, and in its further 12-week extension (KODIAC-08) [20,21].

Recently, two single-country real world studies demonstrating the efficacy and safety of naloxegol in cancer patients with OIC have been published [22,23]. However, data regarding the use of naloxegol in this group of patients in daily clinical practice across Europe do not exist.

We conducted the NACASY study, a multinational European study with the aim to evaluate the safety and efficacy of naloxegol in a real-world setting in patients with cancer pain diagnosed with OIC, during a 4-week follow-up period.

## 2. Material and Methods

### 2.1. Study Design

This was a four-week, single-arm, open-label, multinational, multicenter, prospective, real-world observational study in adult subjects with OIC, receiving naloxegol in routine clinical practice, as prescribed by their physicians according to the conditions established in its Summary of Product Characteristics (SmPC) and with the further recommendation to halt all currently used maintenance laxative therapy.

A total of twenty-six European hospitals participated in the study. The study was conducted according to the requirements expressed in the Declaration of Helsinki [24], abiding to Good Epidemiological Practices [25] and current European regulations relating to the conduct of observational studies. The study was reviewed and approved by the Ethics Committees from the participating sites according to specific local regulations. Written informed consent was obtained from all subjects before enrollment in the study. ClinicalTrials.gov (accessed on 25 November 2021) Identifier: NCT03638440.

### 2.2. Study Population

Patients were eligible for the study if they met all the following inclusion criteria: adult patients (≥18 years of age) with cancer pain, who had been receiving treatment with opioids for at least 4 weeks, were expected to remain on opioids for the entire duration of the study, and had been diagnosed with OIC (defined as <3 documented spontaneous bowel movements [SBMs] per week on average within the previous 2 weeks). In addition, patients must have reported ≥2 of the following symptoms in at least 25% of the bowel movements (BMs) during that period: lumpy (Type 2) or hard (Type 1) stools, according to the Bristol Stool Scale (BSS); straining; sensation of incomplete BM; sensation of anorectal obstruction or blockage; a need for manual maneuvers to facilitate BMs; and, finally, loose stools rarely present without the use of laxatives. These criteria are consistent with the Rome IV diagnostic criteria for OIC [26,27].

Patients were excluded from the study if they were diagnosed with colorectal cancer.

### 2.3. Study Outcomes and Assessments

All the study data were obtained from routine clinical records, from information collected in the diaries provided to the enrolled subjects, or from the selected study questionnaires, and transcribed onto an anonymous case report form (CRF), which was divided into three follow-up sections: Visit 1 (baseline information); Visit 2 (Week 2); and Visit 3 (end of study: at Week 4 after naloxegol initiation).

The following variables were collected along the course of the study: (1) demographic data, cancer and pain clinical characteristics, and treatments; (2) OIC symptoms, laxative treatments, and BMs; (3) naloxegol treatment: initial dose and dose modifications throughout the study follow-up period; (4) all adverse events (AEs), whether related or not to naloxegol treatment, registered and coded according to MedDRA preferred terms (PTs), with their severity consequently graded using the National Cancer Institute Common Terminology Criteria for Adverse Events (CTCAE) v4.03; (5) laxative rescue medication; and 6) the following study questionnaires: (a) the Bowel Function Index (BFI), a physician-administered questionnaire consisting of a three-item patient-assessment scale (ease of defecation, feeling of incomplete bowel evacuation, and personal judgement of constipation); (b) the BSS assessing the stool consistency at the time of every single BM, classifying them into 7 categories (from 1 indicating small, hard, lumpy stool, to 7 denoting watery stool); (c) straining perception assessed by means of a single question and rated according to a 5-point Likert scale (not at all, a little bit, a moderate amount, a great deal, and an extreme amount) and completeness of stool evacuation sensation evaluated by means of a single closed (yes/no) question in the patient’s diary; (d) the Patient Assessment of Constipation—Quality of Life Questionnaire (PAC-QOL), a 28-item self-report questionnaire designed to evaluate the burden of constipation on patients’ everyday HRQoL, covering the specific constipation-related domains of worries and concerns, physical discomfort, psychosocial discomfort, and satisfaction; each symptom’s severity is referred to the 2 previous weeks and scored on a 5-point Likert scale, ranging from 0 (not at all) to 4 (extremely); and (e) the Global Patient Impression for Improvement Questionnaire (PGI-I), consisting of one question for the overall self-assessment of constipation improvement using a 7-point Likert scale (from 1—very much better, to 7—very much worse).

The objective of this study was to assess the safety and efficacy of naloxegol in a real-world setting in cancer patients. The primary safety endpoint was the incidence of adverse events leading to study discontinuation. The primary efficacy endpoint was the response rate during the 4 weeks of treatment. Response rate was defined as the proportion of participants reporting ≥3 BM (without the use of rescue laxative treatment in the previous 24 h) per week and an increase of ≥1 BM over baseline.

Additionally, the following secondary endpoints were evaluated: (a) proportion of patients who had a change in their BFI score of ≥12 points at the end of the study treatment (4 weeks)—this constitutes a clinically important improvement in OIC; (b) proportion of patients who had a BFI score of <30 at the end of the study—this constitutes well-controlled OIC; (c) time to the first post-dose BM; (d) change in stool consistency, according to the BSS [28]; (e) change in the Patient Assessment of Constipation—Quality of Life Questionnaire (PAC-QOL) and the four subscales (physical discomfort, psychosocial discomfort, worries and concerns, and satisfaction) [29]; (f) incidence of overall adverse events, including serious adverse events (SAEs); (g) analgesic treatment interruptions and dose adjustments; (h) naloxegol treatment interruptions and dose adjustments; and (i) patient satisfaction, assessed via the Patient Global Impression for Improvement Questionnaire (PGI-I).

### 2.4. Statistical Analysis

The study sample size was calculated based on the primary safety objective (the incidence of adverse events leading to study discontinuation). Previous studies showed that the incidence of adverse events leading to study discontinuation was about 10% after a follow-up of 12 weeks [19]. Therefore, an overall sample size of 315 patients would be necessary to detect 2% of discontinuations due to adverse events at 4 weeks, with a 95% confidence interval and a precision of ±1.5%.

Exploratory and descriptive methods were used to describe every study variable. All descriptive variables were tabulated. Continuous variables were described by the mean, median, standard deviation, minimum, maximum, and quartiles (range), and categorical variables were presented as distributions of frequencies and percentages. Comparisons of categorical variables were made using the Chi-square or Fisher’s exact test, and for the comparison of quantitative variables, the paired-sample T-test was used. Besides this, associations between different study variables were analyzed using either the T-test for equality of means or the ANOVA test.

For the safety analysis, the population for the analyses comprised all patients who met all selection criteria and had received at least 1 dose of the study drug, while for the efficacy analysis, the population comprised all patients who met all selection criteria, had received at least 1 dose of the study drug, and had at least one post-baseline efficacy assessment. Specifically, two efficacy analyses were performed, based on the days of diary completion: a first one for those subjects who had completed both 4 weeks of treatment and 28 days of the diary (Efficacy Population 1), and a second one for those who had completed at least 21 days of the diary and were not study discontinuations (Efficacy Population 2).

All the data were analyzed using IBM SPSS Statistics version 22.0.

## 3. Results

From August 2018 to January 2020, 183 patients were screened, and 170 recruited by 26 centers from 10 European countries received at least one dose of naloxegol and were included in the safety population analysis. Participant centers included pain units, oncology departments, and palliative care units. Of the 170 patients, 143 patients who had at least one post-baseline efficacy assessment were included in the efficacy analysis; 76 patients were included in Efficacy Population 1 and 98 in Efficacy Population 2 (Figure 1). Overall, there were 56 (32.9%) study discontinuations (Appendix A).

The baseline clinical and demographic characteristics of the patients included in the efficacy population are shown in Table 1. The patients had a median (interquartile range (IQR)) age of 66 years (58–72) and were almost evenly distributed regarding gender. The most frequent primary tumor locations were lung (*n* = 35, 24.5%) and breast (*n* = 32, 22.4%), and 99 (69.2%) patients presented metastasis. The most frequent opioids were fentanyl (*n* = 38, 26,6%), oxycodone (*n* = 36, 25,2%), and morphine (*n* = 16, 16.8%). Baseline opioid therapy was modified at week 2 in 31 of 140 patients (22.1%), with dose increase, dose reduction, and opioid treatment change, in 20, 8, and 3 subjects, respectively; likewise, it was subsequently modified at week 4 in 22 of 118 (18.6%) patients, with dose increase, dose reduction, and opioid treatment change, in 14, 4, and 4 subjects, respectively (Appendix A).

At baseline, 104 of 143 (72.7%) subjects were receiving conventional laxatives, and the corresponding proportions at Visits 2 and 3 were 105/140 (75%) and 43/118 (36.4%), respectively. Osmotic and stimulant laxatives were the most frequently used at every study visit (data not shown). Furthermore, the most frequently used baseline laxatives were: osmotic laxatives (79.8%), stimulant laxatives (30.8%), stool softeners (11.58%), and other (18.3%, enema being the most common, with 9.6%).

Naloxegol 25 mg daily was the initial treatment dose in 139 of 170 (81.8%) patients, 30 (17.6%) patients initiated treatment with naloxegol 12,5 mg daily, and 1 patient with naloxegol 50 mg daily. At Visit 2, 14/140 (10%) patients required a treatment interruption, and 4/140 (2.8%) required a dose increase; the corresponding figures at Visit 3 were 11/118 (7.8%) treatment interruptions and 6/118 (4.3%) dose adjustments (the dose was increased in 5 patients, while the sixth patient had both an increased and a reduced dose adjustment).

## 4. Safety

Overall, 89 of 170 study subjects (52.4%) reported at least one adverse event, and 38 (22.4%) were recorded as serious adverse events. The most frequent (i.e., adverse events with an incidence rate of ≥5%) are presented in Appendix A. There were two cases of withdrawal syndrome that were categorized as grade 1.

Adverse events considered to be related to naloxegol were reported in 23 patients (13.5%). Treatment-related adverse events were mainly gastrointestinal events, the most frequently reported being abdominal pain (*n* = 14, 8.2%) and diarrhea (*n* = 5, 2.9%). Except for a case of intestinal perforation that was categorized as grade 5, the remaining evaluable adverse events were graded 1 to 3 (Table 2). There were two (1.2%) treatment-related adverse events that were considered serious adverse reactions: a case of grade 5 intestinal perforation and a case of grade 2 diarrhea. The intestinal perforation occurred in a 68 years-old male, diagnosed with advanced pancreatic cancer, and with a medical history of gastric bypass surgery. Shortly after initiating treatment with 25 mg of naloxegol, the patient experienced rapid deterioration of the general state, showed signs of peritonitis, sepsis and multiorgan failure, and finally died. Autopsy was not performed, and the event was categorized by the investigator as probable intestinal perforation.

Out of 170 patients, 20 (11.8%, 95%CI 6.9–16.6) discontinued the study due to adverse events, and, of them, 12 (7.1%, 95%CI 3.2–10.9%) were study discontinuations due to naloxegol-related adverse events. These were mainly gastrointestinal side effects, including eight cases of abdominal pain, two cases of diarrhea, and one due to intestinal perforation; the remaining patient discontinued the study due to fatigue.

## 5. Efficacy

### 5.1. Bowel Movements

A total of fifty-five patients (72.4%, 95% CI 62.3%−82.4%) from Population 1 (N = 76) met the primary efficacy endpoint after the 4-week study follow-up period, and they were regarded as responders to naloxegol treatment. After the 4-week study period, the average number of weekly BMs in Population 1 was 6.9 (95% CI, [6.1–7.7]), split into 8.1 (95%, CI (7.3–8.9)) and 3.7 (95% CI, (3.2–4.2)) for the responders and non-responders, respectively (*p* < 0.001) (Figure 2). Likewise, 74 patients (75.5%, 95%CI 67.0%−84.0%) from Population 2 (*n* = 98) responded to naloxegol treatment. The results regarding the number of weekly BMs in Population 2 almost overlapped those of Population 1 (Figure 2).

BM, bowel movement.

Vertical bars represent 95% confidence intervals.

Population 1: Subjects who had completed both 4 weeks of treatment and 28 days of the diary.

Population 2: Subjects who had completed at least 21 days of the diary.

The response to naloxegol was mainly observed within the first week of treatment. The mean number (standard deviation, SD) of BMs per week increased from <3 at baseline to 7.6 (5.0) and 7.3 (4.8) at Week 1 in Populations 1 and 2, respectively. There were no statistically significant differences in the average number of weekly BMs between Weeks 1 and 4 in both populations. The mean (SD) time to the first post-dose bowel movement was 1.9 (1.7) days.

### 5.2. Other Efficacy Outcomes

Of 117 evaluable patients, 75 (64.1%, 95% CI 55.4–72.8%) had a BFI score change of ≥12 points at the end of the study. Likewise, 43 patients (36.8%, 95% CI 28.1–45.5%) had a BFI score of <30 at the study end.

Stool consistency as assessed via the BSS increased by a mean of 0.8 (1.8) points by Week 2 and 0.9 (1.8) points by Week 4. Pairwise comparison showed a significant improvement in the BSS score from baseline to Week 4 (*p* < 0.001); (detailed information on the BSS results is presented in Appendix A).

The number (percentage) of patients reporting a great deal or an extreme amount of straining during bowel movement decreased from 67 out of 135 evaluable patients (49.6%) at baseline to 22 out of 110 evaluable patients (20.0%) at Week 4 (Figure 3). A significant improvement in self-reported straining perception was observed from baseline to Week 2 (*p* = <0.001) and to Week 4 (*p* < 0.001) according to pairwise comparisons.

The number and proportion of patients with sensation of incomplete evacuation decreased from 109 of 137 evaluable subjects (79.6%) at baseline to 80 of 132 subjects (60.6%) at Week 2 and to 63 of 113 subjects (55.6%) at the end of Week 4 of treatment. Based on the pairwise comparisons, the difference in the proportions of patients with sensation of incomplete evacuation between baseline and Week 2 was statistically significant (*p* = 0.007), while that between baseline and Week 4 did not reach statistical significance (*p* = 0.089) (data not shown).

PAC-QOL total score and all its subscales improved from baseline to 4 weeks of follow up. The proportion of patients with clinically relevant improvement in the total PAC-QOL score was 37.1% (*n* = 124). The mean for baseline, Week 2 and Week 4 for the PAC-QOL total score and subscale scores are displayed in Figure 4. All the changes in the PAC-QOL scores reached statistical significance in the pairwise comparisons.

A total of one hundred and eighteen patients (75.0%, 95%CI 67.2–82.8%) reported that their constipation had improved, according to the PGI-I questionnaire, after 4 weeks of treatment. A total of fifty-one (43.2%) patients were “much better” or “very much better”.

Notably, a direct association between the responses to the HrQoL PAC-QOL (total score) and PGI-I questionnaires was observed (*p* < 0.001).

## 6. Discussion

We conducted a multinational (European), multicenter, prospective, real-world observational study aimed to evaluate the efficacy and safety of naloxegol for the treatment of OIC in cancer patients after a 4-week follow-up period. The results from our study support the generally good tolerability and safety profile of naloxegol and its efficacy in patients with cancer-related pain and OIC treated under routine clinical practice conditions. Furthermore, based on these data the efficacy of naloxegol appears to be independent of the type of opioid and relieves OIC quickly.

There was a small proportion of subjects who reported adverse reactions due to naloxegol treatment (13.5%), and only 7.1% of the whole safety population discontinued early from the study as a consequence of naloxegol-related adverse events. This proportion is slightly lower than the 10% reported in phase III, 12-week clinical trials [19], and is within the range reported in similar real-world prospective observational studies: 4.8% after 12 months but all of them within the first 4 weeks [22], and 6.8% after 4 weeks [23], respectively. It is worth mentioning that most of these adverse events frequently occurred within the first two weeks of treatment with naloxegol [20,22]; therefore, it is also possible that most adverse reactions leading to drug discontinuation occur early during treatment. The tolerability profile was consistent with those reported in both clinical trials [19,20] and observational studies [22,23], the most commonly reported naloxegol-emergent adverse events in our study being of gastrointestinal nature and of mild to moderate severity.

Regarding the primary efficacy endpoint, a high proportion of study patients (72.4%) responded to naloxegol treatment during the 4-week study follow-up period. This result is also consistent with the reported percentages from two other similar observational studies with naloxegol, in which the response rate was assessed by the number of BMs over the previous weeks [22,23].

In a 12-month follow-up observational study (Kyonal Study), Cobo et al. showed the sustained efficacy and safety of naloxegol improving OIC, quality of life as measured with the PAC-QOL, and symptoms of cancer patients [22]. A more recent 4-week follow-up study (MovE Study), conducted in France in patients with cancer and OIC, receiving naloxegol treatment also showed high response rates to naloxegol, with an improvement of constipation that was associated with an improvement of quality of life [23]. After 4 weeks of treatment with naloxegol, Cobo et al. in the Kyonal Study reported a response rate of 74.6% [22], and Lemaire et al. in the MovE Study found a response rate of 73.4% [23]. These results are better than those reported in the phase III trials, where investigators observed response rates of 48.7% and 46.8% at Week 12 in the KODIAC-04 and KODIAC-05 studies, respectively [19]. However, our definition of response (i.e., ≥3 SBM per week, without the use of rescue laxative treatment in the previous 24 h, and an increase of ≥1 SBM over baseline) differs from the definition used in clinical trials (i.e., ≥3 SBM per week and an increase from baseline of ≥1 SBM for ≥9 of 12 weeks and for ≥3 of the final 4 weeks). Therefore, the response rates with naloxegol from observational studies and clinical trials are hardly comparable.

In our study, we consistently found significant improvements in most of the secondary efficacy endpoints, despite the short study duration. Statistically (and, more importantly, clinically) significant improvements at the end of the study were observed in the self-perception problems associated with constipation (as measured via the BFI questionnaire), stool consistency (as evaluated through the BSS), and self-reported straining perception. Furthermore, the quality of life of the study subjects had also improved by the end of the study, as reflected by the changes in the PAC-QOL total score and all its subscale scores. This was further corroborated by the significant improvement in the self-perception of efficacy according to the PGI-I questionnaire and by the finding of a significant association between the PAC-QOL total score and PGI-I.

The NACASY study is the first multinational European study evaluating naloxegol for the treatment of OIC in cancer patients in a real-world setting. The limitations of this study are those stemming from its noninterventional design and the absence of a control group, so our findings should be considered supportive of those reported in randomized controlled trials. Missing data in observational studies are an issue and could be responsible for an overestimation of treatment effects compared to the intention-to-treat analysis of the randomized controlled trials. We did not achieve the sample size foreseen in the protocol, increasing the imprecision of our results.

## 7. Conclusions

The findings from the NACASY study support and provide new evidence about the beneficial effect of naloxegol in terms of improvement of constipation and quality of life in patients with cancer-related pain and opioid-induced constipation, with a tolerability and safety profile consistent with that reported by Cobo et al. and Lemaire et al. in the Kyonal and MovE observational studies in patients with cancer, respectively, as well as those reported in phase III clinical trials in patients with noncancer pain. Currently, and despite its high prevalence, OIC remains under-recognized and undertreated, thus impacting on wellbeing and quality of life of cancer patients [30]. In this study, the authors highlight the importance of using precise and practical tools for an efficient OIC diagnosis and management. Taken together, the potential occurrence of OIC should be considered from the start of opioid therapy in cancer patients, allowing patients to receive the right treatment at the right time, thus improving treatment outcomes in this population.

## Figures and Tables

**Figure 1 cancers-14-01128-f001:**
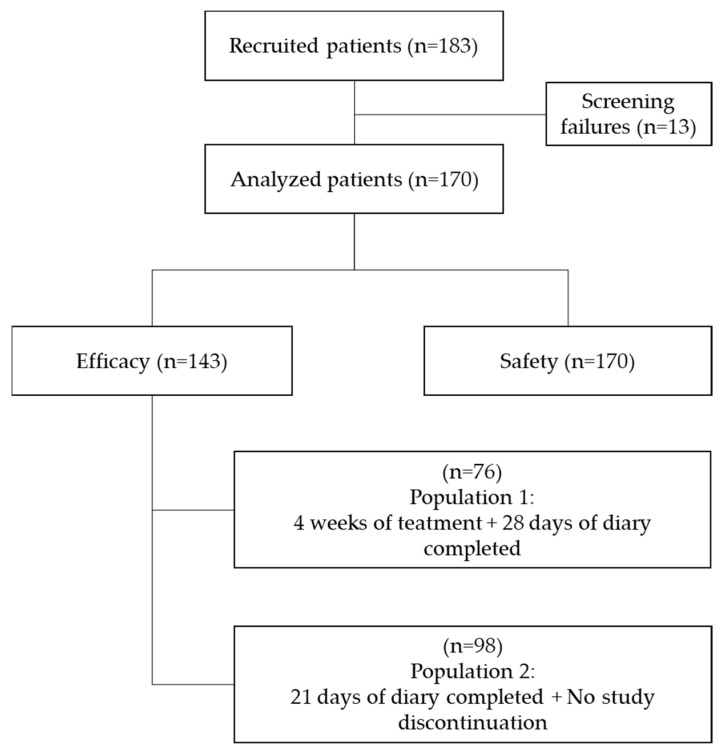
Disposition of subjects.

**Figure 2 cancers-14-01128-f002:**
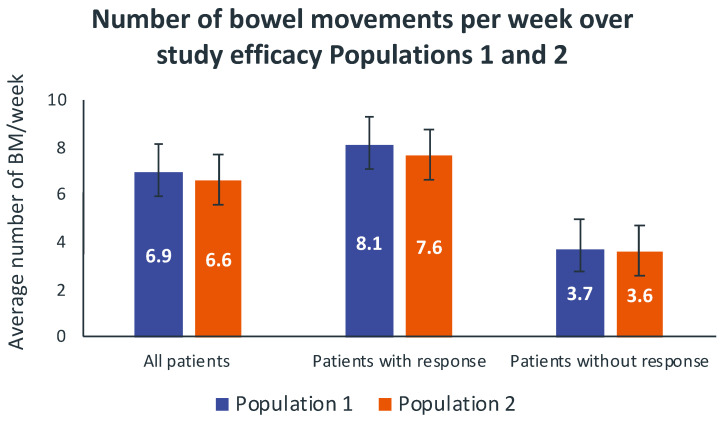
Average number of bowel movements per week over study efficacy Populations 1 and 2.

**Figure 3 cancers-14-01128-f003:**
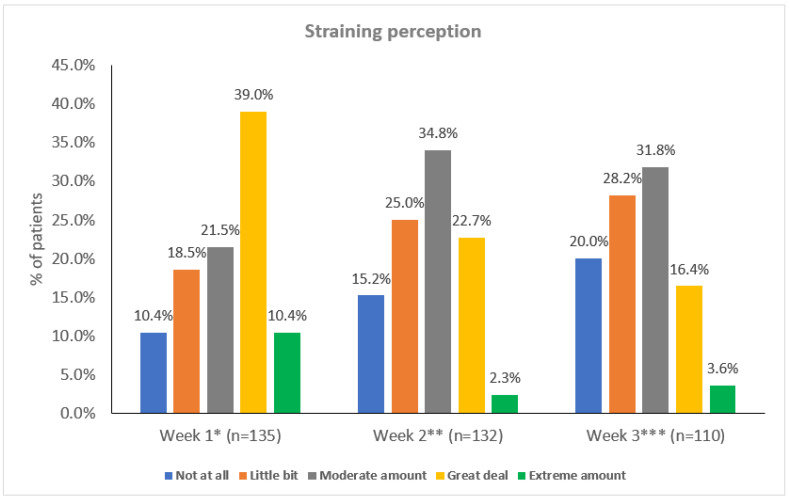
Straining perception during the study. * 8 missing data. ** 11 missing data. *** 9 missing data. Note: 24 patients did not complete Visit 3.

**Figure 4 cancers-14-01128-f004:**
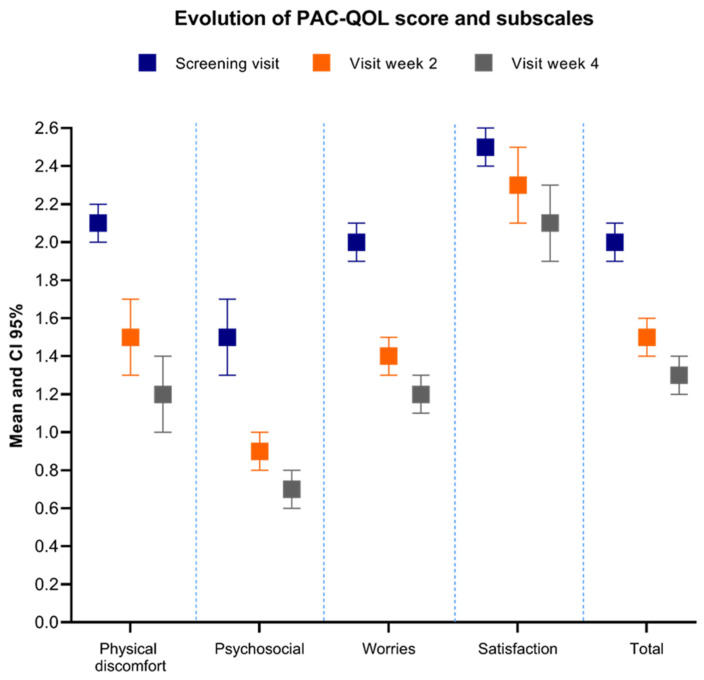
PAC-QOL changes throughout the study.

**Table 1 cancers-14-01128-t001:** Baseline clinical and demographic characteristics.

Baseline Characteristics	Number Of Analyzed Patients	Value
Age, years, median (IQR)	143	66.0 (58.0; 72.0)
Gender, female, N (%)	143	78 (54.5)
Race, Caucasian, N (%)	143	116 (81.1)
Height, cm, median (IQR)	143	165.5 (160.0; 171.0)
Weight, kg, median (IQR)	143	66.0 (58.0; 78.0)
Primary tumor location, N (%)	143	
Lung	143	35 (24.5)
Breast	143	32 (22.4)
Prostate	143	12 (8.4)
Pancreas	143	10 (7.0)
Bladder	143	6 (4.2)
Head and neck	143	5 (3.5)
Uterus	143	4 (2.8)
Bone	143	3 (2.1)
Kidney	143	3 (2.1)
Other	143	33 (23.0)
Presence of metastasis, yes, N (%)	143	99 (69.2)
Current chemotherapy treatment, yes, N (%)	143	66 (46.2)
Carboplatin + Gemcitabine	66	5 (7.6)
Nab-Paclitaxel + Gemcitabine	66	4 (6.1)
Carboplatin + Paclitaxel	66	3 (4.5)
Docetaxel	66	3 (4.5)
Carboplatin + Etoposide	66	2 (3.0)
Opioid treatment at Visit 1, N (%)		
Fentanyl	143	38 (26.6)
Oxycodone	143	36 (25.2)
Morphine	143	16 (11.2)
Codeine *	143	11 (7.7)
Oxycodone/Naloxone	143	10 (7.0)
Hydromorphone	143	8 (5.6)
Methadone	143	7 (4.9)
Tramadol	143	6 (4.2)
Fentanyl + Morphine	143	2 (1.4)
Tramadol/paracetamol	143	2 (1.4)
Tapentadol	143	1 (0.7)
Other	143	6 (4.2)
Previous laxative treatment at Visit 1, yes, N (%)	143	104 (72.7)
Previous laxative treatments at Visit 1, N (%) **		
Osmotic	104	83 (79.8)
Stimulant	104	32 (30.8)
Stool softeners	104	12 (11.5)
Bulking agents	104	4 (3.8)
Linaclotide	104	1 (1.0)
Other	104	19 (18.3)

IQR: Interquartile range. * It includes combinations of codeine plus paracetamol with or without caffeine. ** Percentages have been calculated over the total number of patients with previous laxative treatment (*n* = 104). Patients could receive more than one laxative treatment.

**Table 2 cancers-14-01128-t002:** Adverse reactions to naloxegol (according to CTCAE v4.03).

Adverse Reaction	Grade 1–3		Grade 4–5		Grade NA		Total	
	N	%	N	%	N	%	N	%
Abdominal pain	10	5.9	0	0.0	4	2.4	14	8.3
Diarrhea	4	2.4	0	0.0	1	0.6	5	2.9
Fatigue	1	0.6	0	0.0	0	0.0	1	0.6
Flatulence	2	1.2	0	0.0	0	0.0	2	1.2
Gastrointestinal pain	1	0.6	0	0.0	0	0.0	1	0.6
Intestinal perforation	0	0.0	1	0.6	0	0.0	1	0.6
Nausea	2	1.2	0	0.0	0	0.0	2	1.2
Pollakiuria	1	0.6	0	0.0	0	0.0	1	0.6
Vertigo	1	0.6	0	0.0	0	0.0	1	0.6
Withdrawal syndrome	1	0.6	0	0.0	0	0.0	1	0.6

NA, not available; CTCAE: National Cancer Institute Common Terminology Criteria for Adverse Events. Grade 1: Mild; asymptomatic or mild symptoms; Grade 2: Moderate; minimal, local, or noninvasive; intervention indicated; Grade 3: Severe or medically significant but not immediately life-threatening; Grade 4: Life-threatening consequences; urgent intervention indicated; Grade 5: Death related to AE.

## Data Availability

Data will be available on request due to restrictions e.g., privacy or ethical.

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
