# Peer review of "A Prospective, Real-World, Multinational Study of Naloxegol for Patients with Cancer Pain Diagnosed with Opioid-Induced Constipation—The NACASY Study"

_cancers, 2022, doi:10.3390/cancers14051128_

Round 1

Reviewer 1 Report

The authors present a very interesting prospective multicentric study on the safety and efficacy of naloxegol in patients with cancer pain and cancer-induced constipation. The study tackles an important topic and is well designed and written.  I have only minor recommendations/questions:
  • I think that more information about setting are needed. How many patients were in hospital/hospice ad how many were assisted at home? 
  • There is no description of how and where patients were recruited.
  • Who administered the questionnaires and collected the data?
  • I can not fully understand the flow chart shown in figure 1. Does efficacy population 2 include efficacy population 1? The sum of the two groups is 174 so I assume that some patients are included in both populations.
  • What is the meaning of "No study" in the last part of figure 1 regarding population 2?
  • Line 231: the authors wrote that the initial treatment naloxegol dose was 25 mg/die for 139 patients, what is the dose for the remaining patients? Did you observe differences in the adverse events on the basis of the initial naloxegol dose?
  • I think that there is a mistake in line 321, "...80 of 12 subjects (60.6%)...". Please check.

Author Response

Reviewer 1

The authors present a very interesting prospective multicentric study on the safety and efficacy of naloxegol in patients with cancer pain and cancer-induced constipation.

The study tackles an important topic and is well designed and written.  I have only minor recommendations/questions:

I think that more information about setting are needed. How many patients were in hospital/hospice ad how many were assisted at home?

Most patients were recruited at hospitals, but we have no information on whether they were treated in an out- or inpatient basis.

There is no description of how and where patients were recruited.

Following the reviewer’s suggestion, we have added some information on this regard. Currently, the description reads as follows: “From August 2018 to January 2020, 183 patients were screened, and 170 recruited by 26 centers from 10 European countries received at least one dose of naloxegol and were included in the safety population analysis. Participant centers included pain units, oncology departments and palliative care units”.

Who administered the questionnaires and collected the data?

Except for the Bowel Function Index (BFI) that is a physician-administered tool, the remaining outcome measures were self-administered, including the Patient Assessment of Constipation—Quality of Life Questionnaire. We have clarified this issue for the BFI in the manuscript.

I cannot fully understand the flow chart shown in figure 1. Does efficacy population 2 include efficacy population 1? The sum of the two groups is 174 so I assume that some patients are included in both populations.

As we mention in the statistical analysis section “two efficacy analyses were performed, based on the days of diary completion: a first one for those subjects who had completed both 4 weeks of treatment and 28 days of the diary (Efficacy Population 1), and a second one for those who had completed at least 21 days of the diary (Efficacy Population 2). We have corrected an inconsistency between the figure and the text. These populations were used for the diary-based analyses.

What is the meaning of "No study" in the last part of figure 1 regarding population 2?

There was a problem with the figure (some information was hidden) that has been corrected. The complete information is “No study discontinuation”.

Line 231: the authors wrote that the initial treatment naloxegol dose was 25 mg/die for 139 patients, what is the dose for the remaining patients? Did you observe differences in the adverse events on the basis of the initial naloxegol dose?.

The vast majority of the remaining patients (n=30) received 12.5 mg/day. This information has been added in the text. Consistently with the reviewer’s suggestion, we have analyzed some tolerability outcomes stratified by the initial dose. The proportion of patients showing ADRs was 14.5% among those receiving 25 mg/day and 10% among those receiving 12.5 mg/day. Regarding study discontinuations due to adverse reactions, there were 12 cases (8.6%) among patients who initiated the treatment with 25 mg/day of naloxegol and 0 cases among those who initiated treatment with 12.5 mg/day. This information is difficult to interpret because the small number of patients in the 12.5 mg/day subgroup. It should bear in mind that this information is based on observational data and using flexible dose and it is not robust or powered enough to make any assumption regarding a dose relationship.

I think that there is a mistake in line 321, “…80 of 12 subjects (60.6%)…”. Please check.

Thank you for detecting this mistake. The figure has been corrected (“80 of 132”)

Reviewer 2 Report

This is an international, multi-center, open label, non-randomized trial of the effect of naloxegol on opioid-induced constipation in patients with neoplastic diseases. The manuscript is well written. The following comments are offered in order to strengthen an already robust presentation.

Title: sufficiently describes the study.

Abstract: include and spell out the NACASY study.

Key words: add 'PAMORA.' 

Introduction: around line 80, please include the typical dose range.

Methods: validated tools and instruments were used, including Bristol and PAC-QOL. Include the dose range available for use. How were the hospitals recruited for participation?

Results: Table 2 first column needs a heading, such as Adverse effects. Otherwise, results are well described and presented. Include the dose range used during the study.

Discussion: uniqueness of this study was highlighted as real-world, international, multi-center and confirmed findings in other trials. Limitations are well described.

Conclusion: could separate conclusion from discussion.

Funding: a statement of the involvement of the study's funder, Kyowa Kirin Pharmaceutical Development Ltd, is necessary. Please include access to data, analysis of findings,  and manuscript development.

References: mostly in mdpi style.

Thank you for the opportunity to review and comment on this manuscript.

Author Response

Reviewer 2

This is an international, multi-center, open label, non-randomized trial of the effect of naloxegol on opioid-induced constipation in patients with neoplastic diseases. The manuscript is well written. The following comments are offered in order to strengthen an already robust presentation.

Title: sufficiently describes the study.

Abstract: include and spell out the NACASY study.

We have included the name of the study

Key words: add ‘PAMORA.’

Together with MOVANTIK, It has been added to the key words                                                                                                                                       

Introduction: around line 80, please include the typical dose range.

It has been added

Methods: validated tools and instruments were used, including Bristol and PAC-QOL. Include the dose range available for use. How were the hospitals recruited for participation?

Although the protocol included the recommendation to follow the Summary of Product Characteristics, because of the study was conducted under real-world practice conditions, physicians prescribed naloxegol according to their clinical judgement.

The participating centers were selected by the sponsor based on the previous experience with these centers

Results: Table 2 first column needs a heading, such as Adverse effects. Otherwise, results are well described and presented. Include the dose range used during the study.

The heading has been added

Discussion: uniqueness of this study was highlighted as real-world, international, multi-center and confirmed findings in other trials. Limitations are well described.

Conclusion: could separate conclusion from discussion.

We have separated conclusion from discussion

Funding: a statement of the involvement of the study's funder, Kyowa Kirin Pharmaceutical Development Ltd, is necessary. Please include access to data, analysis of findings,  and manuscript development.

The statement has been included. It reads as follows: “Role of the funding source:  the funder of the study and its employees and assignees were involved in study design, data collection, data analysis, data interpretation, and writing of all related reports and publications”

References: mostly in mdpi style.

Thank you for the opportunity to review and comment on this manuscript.